# ^1^H-NMR Metabolomics Analysis of the Effects of Sulfated Polysaccharides from Masson Pine Pollen in RAW264.7 Macrophage Cells

**DOI:** 10.3390/molecules24091841

**Published:** 2019-05-13

**Authors:** Fangchen Su, Mengmeng Sun, Yue Geng

**Affiliations:** 1Key Laboratory of Food Nutrition and Safety of SDNU, Provincial Key Laboratory of Animal, Resistant Biology, College of Life Science, Shandong Normal University, Jinan 250014, China; sufangchen0314@163.com; 2State Key Laboratory of Biobased Material and Green Papermaking, Qilu University of Technology, Shandong Academy of Sciences, Jinan 250353, China; sun007_2007@163.com

**Keywords:** ^1^H nuclear magnetic resonance, metabolomics, sulfated polysaccharides

## Abstract

Many polysaccharides have been shown to be bioactive, with the addition of sulfate often enhancing or altering this bioactivity. In previous studies, masson pine pollen polysaccharides, to include a sulfate derivative, have been shown to promote macrophage proliferation similarly to LPS. However, the exact metabolic mechanisms promoting this proliferation remain unclear. In this study, RAW264.7 macrophage cells were treated with a purified masson pine pollen polysaccharide (PPM60-D), a sulfate derivative (SPPM60-D), or LPS. Proliferation levels at a variety of concentrations were examined using MTT assay, with optimal concentration used when performing metabolomic analysis via ^1^H nuclear magnetic resonance (^1^H-NMR). This process resulted in the identification of thirty-five intracellular metabolites. Subsequent multivariate statistical analysis showed that both LPS and SPPM60-D promote RAW264.7 proliferation by promoting aerobic respiration processes and reducing processes associated with glycolysis. While some insight was gained regarding the mechanistic differences between SPPM60-D and LPS, the specific mechanisms governing the effect of SPPM60 on RAW264.7 cells will require further elucidation. These findings show that both LPS and SPPM60-D effectively promote RAW264.7 proliferation and may have beneficial uses in maintaining cellular vitality or inhibiting cancer.

## 1. Introduction

Masson pine pollen polysaccharides serve various functions and have varying degrees of activity. When these polysaccharides are artificially modified, such as with the addition of sulfate, their activity can be improved due to their branched chain now carrying negative charges [1]. This alteration can also impact the bioactivity of the polysaccharide and confer valuable properties, such as antioxidant, antiviral, antitumor, anticoagulant or pro-immunity [2]. In our previous study, we showed that sulfated polysaccharides from masson pine pollen could promote spleen lymphocyte and macrophage proliferation and could induce the generation of immune-related cytokines [3].

The field of metabolomics seeks to identify and quantify metabolites in biological systems. Differential metabolites can be identified by utilizing multivariate statistical analyses, and their identification can aid in early disease diagnosis or facilitate the identification of upstream pathways or corresponding genes. At present, high performance liquid chromatography (HPLC), gas chromatography (GC), mass spectrometry (MS), nuclear magnetic resonance (NMR), and infrared (IR) spectroscopy are commonly used detection approaches [4]. NMR offers high sensitivity, reproducibility, and suitability for detection within mixed samples and in vitro or in vivo experimentation. Furthermore, NMR offers a reduced cost compared with the alternatives and can easily identify metabolite changes; thus, it is an indispensable tool in metabolomic research [5].

Macrophages are a part of innate immunity and play an important role in immune responses by releasing cytokines and signal factors in vivo. While research has been performed to examine the mechanisms of macrophage activation using metabolomic focused technologies [6,7,8], the examination of RAW264.7 macrophage activation by masson pine pollen polysaccharides using NMR has not been previously performed. Polysaccharides have been shown to act on relatively complex pathways, with the addition of sulfate enhancing or sometimes altering these actions. Herein, masson pine pollen polysaccharides or a sulfate derivative were utilized to induce RAW264.7 cell proliferation, with the mechanisms of this outcome examined using the ^1^H-NMR metabolomic approach.

## 2. Material and Methods

### 2.1. Materials and Reagents

Broken masson pine (*Pinus massonian*) pollen was provided by the Yantai New Era Health Industry Company (broken rate < 95%). Broken masson pine pollen polysaccharides were extracted using a water extract-alcohol precipitation method, with the coarse polysaccharides (PPM60) obtained by ethanol precipitation with 60% ethanol. Proteins were removed using the trichloroacetic acid precipitation method. PPM60-D was then purified from PPM60 using Sephacryl S-400HR and was composed of mannose, galactose, trehalose, and an unknown monosaccharide at molar ratios of 0.75:1:0.68:2.37 [9]. A sulfate derivative, namedSPPM60-D, was then obtained using chlorosulfonic- acid pyridine. Ultraviolet spectrum analysis and the barium sulfate turbidity method were used to assess sulfur content, with the degree of substitution then calculated.

RAW264.7 cells were purchased from the Chinese Type Culture Collection (CTCC, Shanghai, China). Dulbecco’s modified Eagle’s medium (DMEM) was obtained from Gibco (New York, NY, USA); fetal calf serum from Zhejiang Tianhang Biotechnology Co. (Huzhou, China); and 100×penicillin-streptomycin, MTT, DMSO, lipopolysaccharide (LPS), and phosphate buffer from Solarbio (Beijing, China). Sephacryl S-400 HR was purchased from Pharmacia (Helsingborg, Sweden); 5 mm ST500-7 NMR sample tubes from Norell (Morganton, NC, USA); and deuterium oxide with 0.05% TSP, Dextran T10, Dextran T70, Dextran T110, and Dextran T500 from Sigma Aldrich (Saint Louis, MO, USA). Other analytical reagents were prepared in-house.

### 2.2. Instrument

The instrumentation used included a TU-1810SPC UV-VIS spectrophotometer (Beijing Purkinje General Instrument co., LTD., Beijing, China), 5804R high-speed refrigerated centrifuge (Eppendorf, Germany), SB-1000 rotary evaporator (EYELA, Tokyo, Japan), FDU-1200 freeze dryer (EYELA, Japan), carbon dioxide incubator (Nuaire, USA),VC 130PB ultrasonic cell disrupter system (Sonic, Branson, MO,, USA), Avance 600 III nuclear magnetic resonance spectrometer (Bruker, Karlsruhe, Germany), and U3000 high performance liquid chromatography with refractive index detector (Diane, Sunnyvale, CA, USA).

### 2.3. Effects of SPPM60-D on RAW264.7 Cell Proliferation

RAW264.7 cell proliferation was assessed in the presence of SPPM60-D, PPM60-D, or LPS (positive control) using MTT assay. RAW264.7 cells were plated at a concentration of 1 × 10^4^ cells/mL in a 96-well plate. Samples were then treated with PPM60-D or SPPM60-D at a final concentration of 100, 200, 400, or 800 μg/mL or with LPS (10 μg/mL) for 48 h. Next, MTT (20 μL) was added to each well and allowed to incubate for 4h. At last, the plate was centrifuged to remove the supernatants, DMSO (150 μL) was added to each well, and measurements were obtained at an absorbance of 490 nm.

### 2.4. ^1^H-NMR Metabolomics Analysis of the Effects of SPPM60-D on RAW264.7 Cells

RAW264.7 cells were plated at a concentration of 1 × 10^4^ cells/mL, treated with either SPPM60-D (400 μg/mL) or LPS (10 μg/mL) for 48 h, and rinsed twice with PBS. A negative control group that was only cultured in DMEM was also used. Cells were quenched using cold methanol and detached using cell lifters. The intracellular fluid was extracted using a dual phase extraction procedure with methanol–chloroform and ultrapure water as previously described [10]. The supernatants were transferred to new vials, with the hydrophilic metabolites obtained after drying and dissolved in 600 μL D_2_O. The ^1^H-NMR measurements were performed on a Bruker Avance 600 III nuclear magnetic resonance spectrometer, with 550 μL of each sample solution added to a 5 mm NMR tube for analysis.

### 2.5. Data Analysis

A MestReNova 6.1.1 (Mestrelab Research SL, Santiago De Compostela, Spain) was used to add exponential window function (exponential: 0.5) and to improve the signal to noise ratio (SNR). Manual phase baseline correction with TSP calibration was used to remove the water peak at 4.65–5.15 ppm and to eliminate the effect of water on the integral. The integral width (bin) was 0.002. The total area of the spectrogram was normalized, and the saved format of the integral data was a .txt file. To open the excel table, manually we input packet information and sample number, remove the water peak section and import the SIMCA-P+12.0 software (Umetrics Inc., Umea, Sweden) for partial least squares discriminant analysis (PLS-DA) and orthogonal partial least squares discriminant analysis (OPLS-DA). The reliability of the PLS and OPLS-DA model was verified by permutation test and CV-ANOVA, and the difference variables were screened. The main criterion was the variable importance in projection (VIP) value and load weight and correlation coefficient (Pcorr) for the first prediction principal component.

## 3. Results

### 3.1. SPPM60-D Characterization

The average molecular weight of SPPM60-D was determined using HPLC. The columns were calibrated using various dextran standards and a calibration curve was derived based on the equation Log (Mw) = −0.2582t + 11.757 (R^2^ = 0.951) and showed SPPM60-D to have a Mw = 40.5 kDa. Furthermore, fourier-transform infrared spectroscopy (FTIR) was performed and showed that SPPM60-D had been successfully sulfated (Figure 1), with the barium sulfate turbidity method showing the degree of substitution to be 1.212.

### 3.2. Effects of PPM60-D and SPPM60-D on RAW264.7 Cell Proliferation

The MTT assay showed that both PPM60-D and SPPM60-D could promote RAW264.7 proliferation in a dose-dependent manner (Figure 2). PPM60-D only showed significant proliferation levels relative to the negative control at a concentration of 800 μg/mL (*p* < 0.05), while SPPM60-D showed significantly increased proliferation levels at all concentrations in a dose-dependent manor (*p* < 0.01). The highest proliferation levels in the SPPM60-D group were obtained at concentrations of 400 and 800 μg/mL, but no significant differences were noted; thus, a concentration of 400 μg/mL was utilized for subsequent experimentation.

### 3.3. Intracellular Metabolites Identified by Nuclear Magnetic Resonance Spectroscopy (NMR)

An ^1^H-NMR spectrum was constructed for the LPS group (Figure 3A) and the SPPM60-D group (Figure 3B), with associated blank control groups. To identify the indicated intracellular metabolites, relevant literature, the human metabolome database (HMDB), and the biological magnetic resonance database (BMRB) were consulted and thirty-five identities were assigned (Table 1). When comparing the LPS and SPPM60-D spectra, no apparent differences were noted, but the concentrations of some of the metabolites differed.

### 3.4. Data Analysis

#### 3.4.1. Supervised Partial Least Squares Discriminate Analysis (PLS-DA)

The PLS-DA results showed that the values of all the distances between sample-points and the X-line were less than the linear value of 1.35 (Figure 4A,B), thus indicating that there were no abnormal points. The PLS-DA score plots showed that sample points from the SPPM60-D and LPS groups and their associated blank control groups were distinguishable at the first principal component (PC1; Figure 5A,B). Furthermore, the substitution permutation test (Figure 6A,B) showed that the model was reliable and had no over fitting. Overall, this approach was shown to be able to reliably predict data, with the four groups of samples being within the 95% confidence interval.

#### 3.4.2. Orthogonal Partial Least Squares Discriminant Analysis (OPLS-DA)

The OPLS-DA score plots showed that the groups were clearly distinguishable (Figure 7A,B). To determine significance, an ANOVA of the cross-validated residuals (CV-ANOVA) was performed with both the LPS (*p* < 0.046) and SPPM60-D (*p* < 0.0018) sets. Then based on the variable importance in the projection (VIP) values, correlation coefficients (r) and OPLS-DA plot diagrams (Figure 8A,B) of differential metabolites were identified (Table 2). While the types of metabolic pathways between the LPS and SPPM60-D groups were very similar, the concentrations of some of the metabolites did differ. When comparing the LPS group with the blank control group, eighteen differential metabolites were identified, while fifteen were identified in the SPPM60-D group relative to the control. These differential metabolites were amino acid, organic acid, choline, alcohol, and other substances involved in the Krebs cycle, anaerobic respiration, and the synthesis and degradation of amino acids.

#### 3.4.3. Analysis of Differential Metabolites

Based on the pattern recognition analysis and the multivariate statistical results, the Metabo Analyst software and the Kyoto Encyclopedia of Genes and Genomes (KEGG) were utilized to analyze pathways associated with the identified differential metabolites (Table 2). The results showed that in the LPS group, the most significant cellular metabolism pathways associated with the differential metabolites included protein synthesis; glycine, serine, and threonine metabolism; methionine metabolism; betaine metabolism; and ammonia recycling, (Figure 9 and Figure 10; Table 3). When analyzing the SPPM60-D group, the most significant pathway associations included glycine, serine, and threonine metabolism; methionine metabolism; arginine and proline metabolism; and betaine metabolism (Figure 11 and Figure 12; Table 4).

## 4. Discussion

In RAW264.7 macrophages, LPS can induce inflammatory reactions. LPS was shown to be involved in macrophage activation and the production of inflammation factors [11]. Furthermore, LPS has been shown to act through the TLR2/TLR4-PLCγ2 pathway and can promote [Ca^2+^]_i_ in macrophages (Aki et al., 2008). Additionally, one study indicated that LPS can induce Ca^2+^ flow into cells by way of Ca^2+^ release-activated Ca^2+^ (CRAC) channels in mice macrophages [12]. In a previous study, Geng et al. examined the PPM60-D sulfated derivative SPPM60-D in RAW264.7 macrophage cells and showed that it also promoted proliferation similar to LPS by activating Ca^2+^ signaling via toll-like receptor 4 (TLR4) and CRAC channels [3]. However, this study did not examine the possible mechanistic differences between LPS and SPPM60-D.

In the present study, metabolite pathway enrichment analysis (MPEA)of the LPS group showed that pathway inductions were focused on protein synthesis and betaine, taurine, hypotaurine, glycine, serine, and threonine metabolism. Most of these metabolites showed increased concentrations, with the exception of taurine and hypotaurine which were decreased. MPEA was also performed for the SPPM60-D group and showed an induction of pathways associated with protein synthesis and methionine, glycine, serine, threonine, betaine, taurine, and hypotaurine metabolism. Thus, similar trends were seen in both LPS-induced and SPPM60-D induced proliferation in RAW264.7 cells.

Both LPS and SPPM60-D exposure resulted in decreased lactic acid levels. Due to the fact the RAW246.7 cells are an immortalized cell line, you would expect them to exhibit similarities to cancer cells, such as obtaining energy predominantly by glycolysis and producing large amounts of lactic acid [13]. These results show that the masson pine pollen polysaccharides and LPS can inhibit glycolysis, thereby potentially preventing cell cancerization. In one study, it was suggested that the cancer cells choosing an inefficient energy process may be due to impaired mitochondria [14]. Therefore, we postulate that the masson pine pollen polysaccharides and LPS may have a protective effect in mitochondria.

Also, in the LPS and SPPM60-D groups, taurine pathways were decreased. Taurine is an abundant essential non-protein amino acid in animals that is widely distributed in various tissues and is not involved in protein synthesis. Taurine can be synthesized in the liver and also can be obtained through diet. Taurine has a wide range of biological functions, such as regulating normal physiological activities, maintaining osmotic pressure balance, modulating cellular calcium balance, improving immune functions, and enhancing the antioxidant capacity of the cell membrane [15]. Cells can also transport the extracellular taurines into cells through transporters, with one study showing that alanine can have an antagonistic action on taurine transport [16]. In agreement with this, both the LPS and SPPM60-D groups showed significantly higher levels than the control group, thus possibly inhibiting taurine transport. Taurine can be synthesized from cysteine and methionine by cysteine sulfinic acid decarboxylase (CSAD), which produces hypotaurine and then taurine via oxidation. Thus, it would seem that the reduction in taurine may be attributed to reduced CSAD levels.

Other similarities seen in the LPS and SPPM60-D groups included increased proline, methionine, and arginine concentrations, with these amino acids being associated with cell viability and cellular metabolism. The concentrations of cholines and choline phosphates, which are important in the process of energy shuttling, were also significantly increased in the two groups relative to the control. Betaine concentrations were also significantly increased, with betaine metabolism mainly occurring in the mitochondria of liver and kidney cells. Betaine can regulate cellular osmotic pressure and in the betaine homocysteine S-methyltransferase (BHMT) pathway, it serves as a methyl donor for homocysteine (Hcy), which is then utilized to generate methionine, a participant in protein and fat metabolism [17]. These results show that LPS and SPPM60-D utilize many of the same mechanisms when promoting RAW246.7 proliferation.

Despite many commonalities being noted following treatment with LPS or SPPM60-D, some differences were also present. For example, in the SPPM60-D group, ethanol concentrations were reduced relative to the blank control, while the LPS group showed higher levels. In macrophages, transient receptor potential vanilloid subtype 1 (TRPVl) expression, which can be activated by ethanol, has been associated with inflammatory responses [18]. TRPV1 is a nonselective calcium ion channel and is associated with heat and pain responses [19]. Furthermore, TRPV4, another member of the TRPV family, has been associated with the regulation of thermogenesis and inflammation in fat cells [20]. Thus, we speculate that during LPS exposure, ethanol may act as an activator of TRPV1, thereby mediating inflammatory related calcium ion cellular influx and reducing the release of inflammatory cytokines. However, in the SPPM60-D group, ethanol levels were reduced and the TRPV1 channels were not activated. These findings were consistent with a previous study that found that SPPM60-D could promote the release of macrophage inflammatory factors to a higher level than could LPS [3]. However, further study is required to determine if this is associated with the metabolic differences between the two groups.

Another differential factor between the groups was trimetlylamine oxide (TMAO), which was increased in the SPPM60-D group and decreased in the LPS group. TMAO can promote protein folding and ligand binding and enhance protein stability [18]. Additionally, TMAO also has physiological and biochemical properties, such as regulating osmotic pressure. These results suggest that SPPM60-D can offer a protective effect by stimulating macrophages to produce TMAO.

Overall, the present study identified thirty-five intracellular metabolites via NMR spectrometry. Multivariate statistical analyses showed that LPS and SPPM60-D can promote RAW264.7 cell proliferation by promoting aerobic respiration processes and reducing glycolysis. While some insight was gained regarding mechanistic differences between SPPM60-D and LPS, the specific mechanisms governing the effect of SPPM60 on RAW264.7 proliferation require further elucidation.

## Figures and Tables

**Figure 1 molecules-24-01841-f001:**
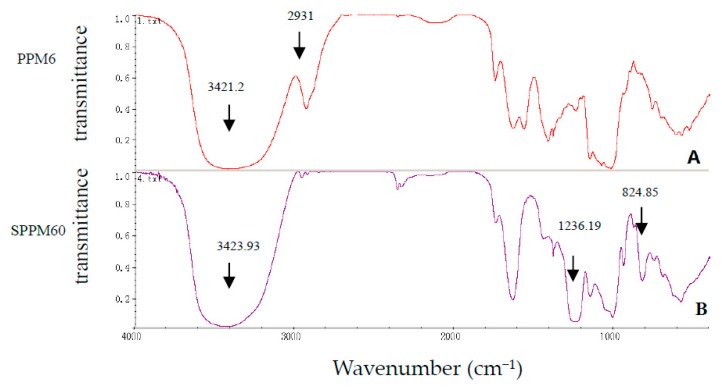
The infrared spectrum of PPM60-D (**A**) and SPPM60-D (**B**).

**Figure 2 molecules-24-01841-f002:**
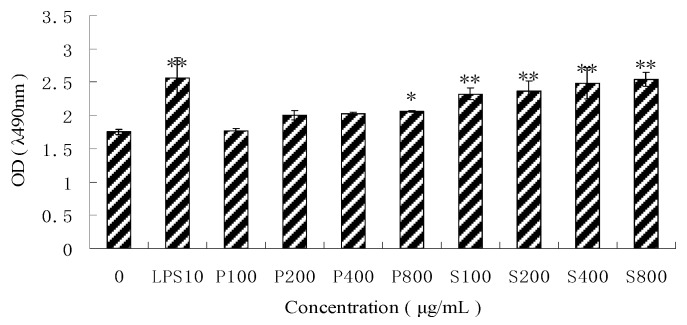
The different concentration of PPM60-D and SPPM60-D effect on RAW264.7 cell proliferation (*n* = 6). Note: * *p* < 0.05, ** *p* < 0.01 compared with the control.

**Figure 3 molecules-24-01841-f003:**
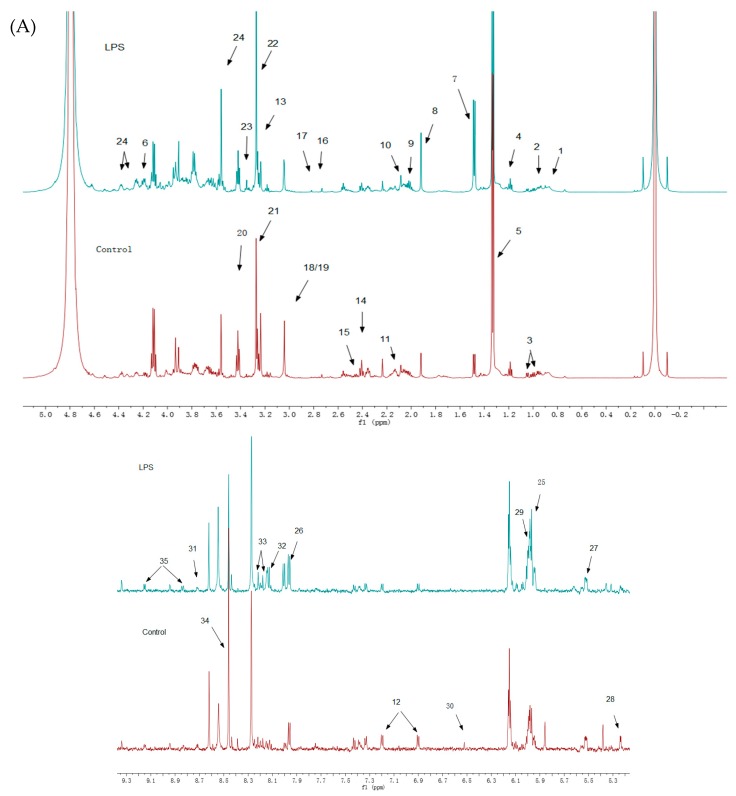
^1^H-NMR spectra of RAW264.7 cell from the LPS groups, SPPM60-D groups and control groups (The arrows show the differential metabolites in Table 1). (**A**) The LPS group vs. Control group (**B**) The SPPM60-D group vs. Control group.

**Figure 4 molecules-24-01841-f004:**
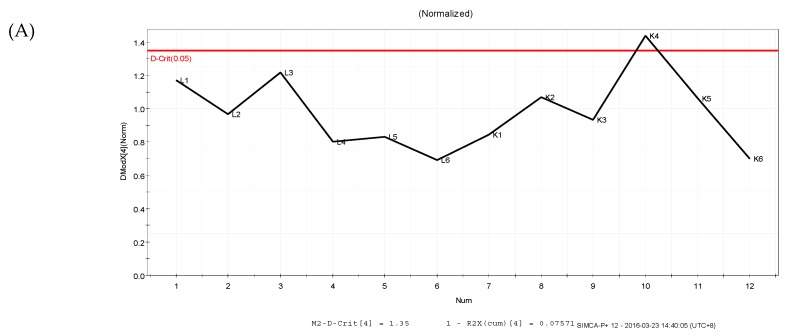
Distance to model X line Plot prediction in samples. (**A**) LPS and control groups; (**B**) SPPM60-D and control groups.

**Figure 5 molecules-24-01841-f005:**
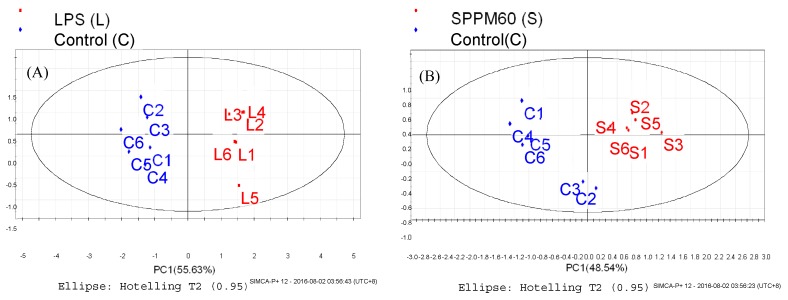
PLS-DA score plot of RAW264.7 cells from treatment and control groups. (**A**) LPS and control groups (**B**) SPPM60-D and control groups.

**Figure 6 molecules-24-01841-f006:**
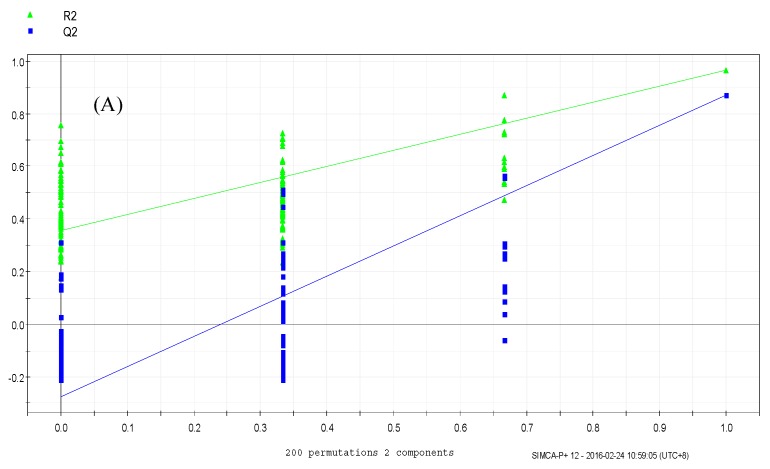
Permutation test of RAW264.7 cells from treatment and control groups. (**A**) LPS and control groups (**B**) SPPM60-Dand control groups.

**Figure 7 molecules-24-01841-f007:**
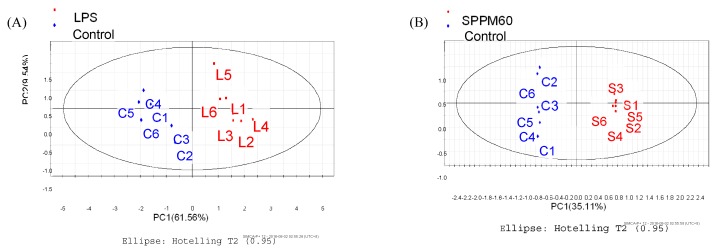
Orthogonal partial least Squares discriminant analysis OPLS-DA score plot of RAW264.7 cells from treatment and control groups. (**A**) LPS and control groups (**B**) SPPM60-Dand control groups.

**Figure 8 molecules-24-01841-f008:**
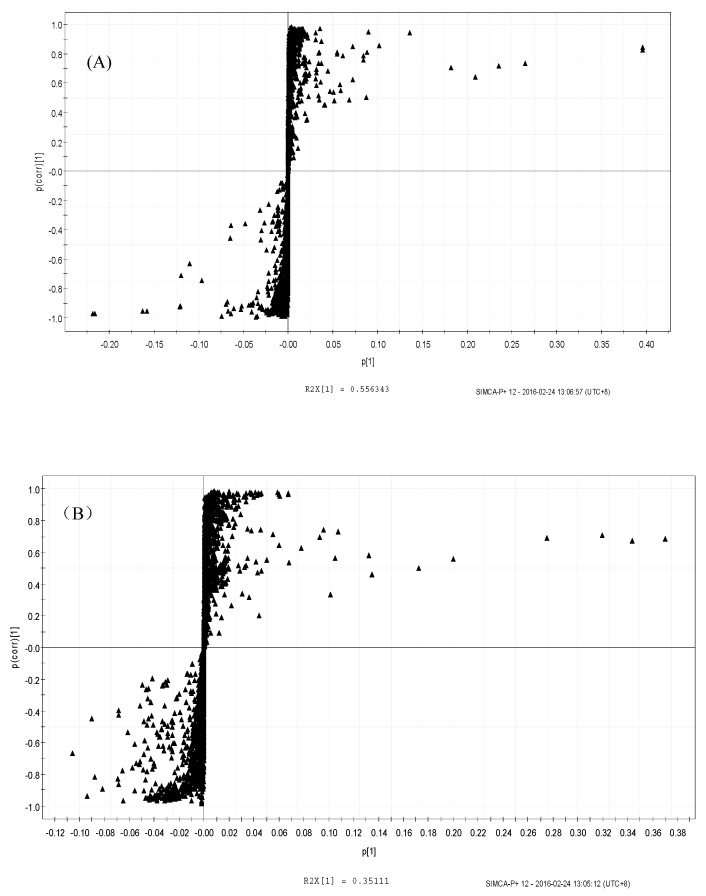
S-Plot of OPLS-DA analysis of RAW264.7 cells from treatment and control groups. (**A**) LPS and control groups (**B**) SPPM60-Dand control groups.

**Figure 9 molecules-24-01841-f009:**
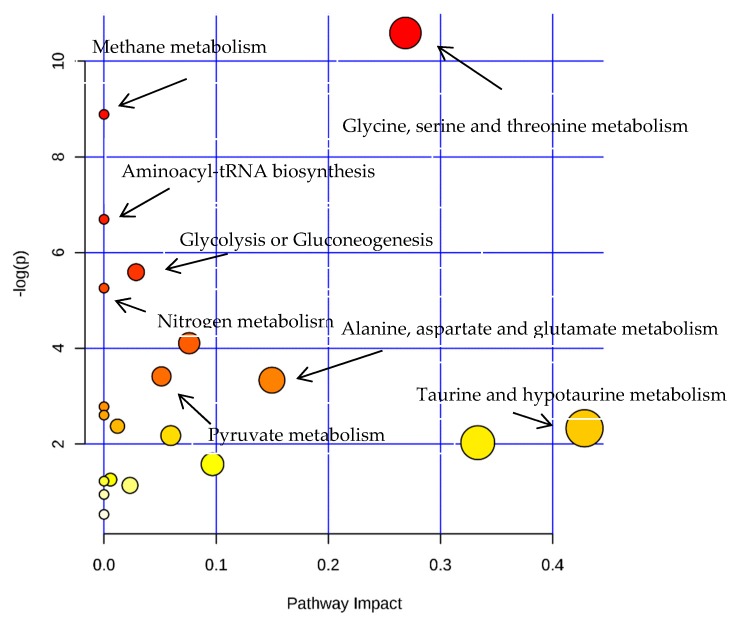
The pathway analysis of all differential metabolites between LPS and control groups.

**Figure 10 molecules-24-01841-f010:**
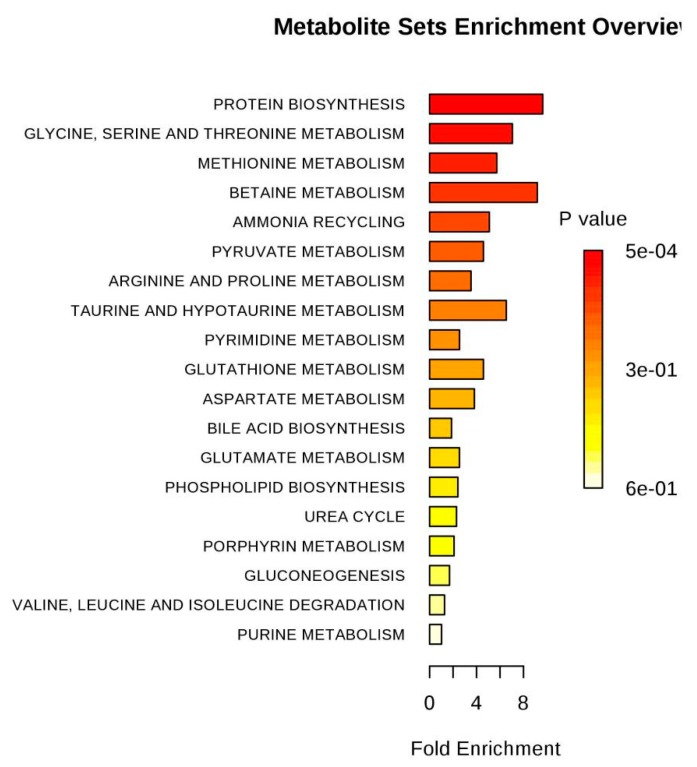
Pathway enrichment of differences metabolites between LPS and control groups.

**Figure 11 molecules-24-01841-f011:**
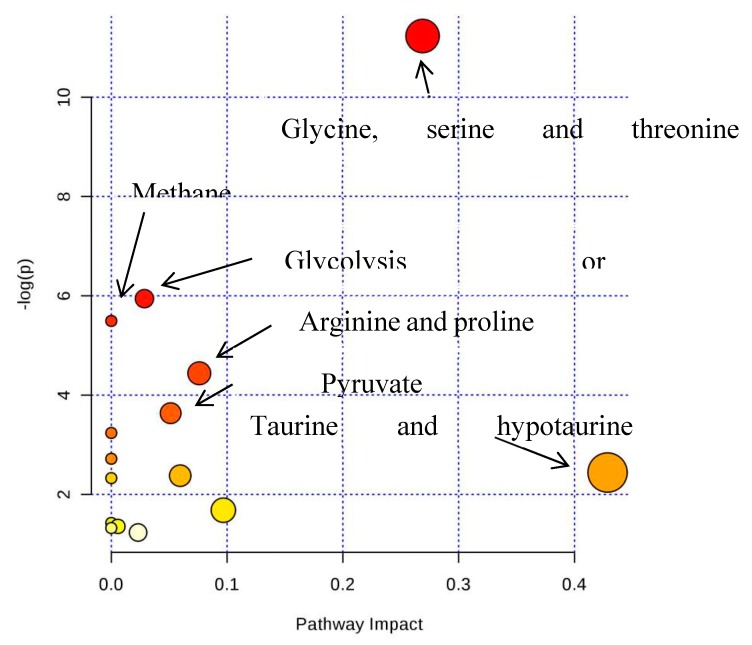
The pathway analysis of all differential metabolites between SPPM60-D and control groups.

**Figure 12 molecules-24-01841-f012:**
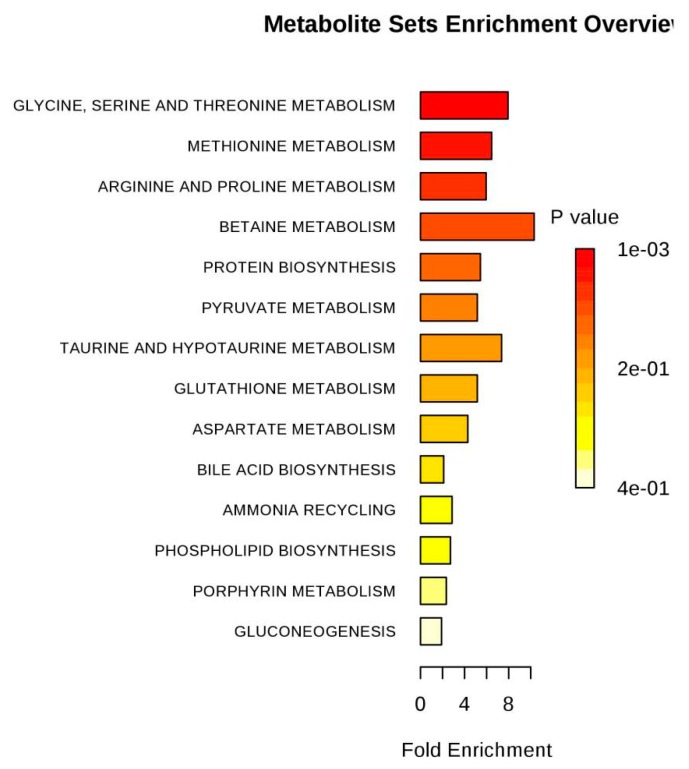
Pathway enrichment of differences metabolites between SPPM60-Dand control groups.

**Table 1 molecules-24-01841-t001:** Nuclear magnetic resonance spectroscopy (NMR) data for RAW264.7 cells.

No.	Metabolites	Group	δH (Multiplicity)	Loading (SPPM60-D vs. Control)	Loading (LPS Group vs. Control)
1	Isoleucine	CH_3_	0.94 (t)	64.9	9.64
		CH_3_	0.99 (d)		
		CH_2_	1.28 (m)		
		CH_2_	1.46 (m)		
		CH	1.99 (m)		
		CH	3.66 (d)		
2	Leucine	CH_3_	0.96 (d)	10.73	22.66
		CH_3_	0.97 (d)		
		CH	1.7 (m)		
		CH_2_	3.77 (m)		
		CH	3.722 (m)		
3	Valine	CH_3_	0.99 (d)	11.7	20.34
		CH_3_	1.05 (d)		
		CH	2.25 (m)		
		CH	3.61 (d)		
4	Ethanol	CH_3_	1.19 (t)	−150.81	146.23
		CH_2_	3.65 (q)		
5	(R)-Lactate	CH_3_	1.33 (d)	−1175	−1020.86
		CH	4.11 (q)		
6	Threonine	CH_3_	1.33 (d)	41.35	40.5984
		CH	3.58 (d),		
		CH	4.25 (m)		
7	Alanine	CH_3_	1.48 (d)	51.01	137.05
		CH	3.78 (q)		
8	Acetate	CH_3_	1.92 (s)	76.87	98.77
9	Proline	CH_2_	2.01 (m)	476.19	22.39
		CH_2_	2.05 (m)		
		CH_2_	2.34 (m)		
		CH_2_	3.42 (td)		
		CH_2_	3.45 (m)		
		CH	4.12 (m)		
10	L-Glutamic acid	CH_2_	2.04 (m)	−15.76	−17.08
		CH_2_	2.11 (m)		
		CH_2_	2.34 (m)		
		CH	3.77 (m)		
11	L-Glutamine	CH_2_	2.13 (m)	−18.93	−35.01
		CH_2_	2.44 (m)		
		CH	3.77 (m)		
12	Tyrosine	CH_2_	3.06 (dd)	2.306	6.27
		CH_2_	3.15 (dd)		
		CH	3.94 (dd)		
		3,5-CH	6.91 (d)		
		2,6-CH	7.2 (d)		
13	Choline	N(CH_3_)_3_	3.23 (s)	306.01	263.07
		N CH_2_	3.56 (s)		
		O CH_2_	4.07 (t)		
14	Succinate	CH_2_	2.393 (s)	10.04	17.12
15	Pyruvate	CH_3_	2.46 (s)	10.48	28.31
16	Sarcosine	CH_3_	2.73 (s)	32.18	11.84
		CH_2_	3.6 (s)		
17	Aspartate	CH_2_	2.68 (m)	67.57	118.274
		CH_2_	2.82 (m)		
		CH	3.91 (m)		
18	Phosphocreatine	CH_3_	3.05 (s)	201.21	142.58
		CH_2_	3.95 (s)		
19	Creatine	CH_3_	3.04 (s)	259.98	226.48
		CH_2_	3.94 (s)		
20	Taurine	CH_2_SO_3_	3.25 (t)	−304.83	−244.785
		NCH_2_	3.43 (t)		
21	Trimethylamine N-oxide (TMAO)	CH_3_	3.26 (s)	54.23	−189.45
22	Betaine	CH_3_	3.27 (s)	294.88	529.127
		CH_2_	3.89 (s)		
23	Methanol	CH_3_	3.352 (s)	−14.48	29.20
24	Glycine	CH_2_	3.56 (s)	306.01	263.07
25	Uridine	CH_2_	3.81 (d)	23.93	118.274
		CH_2_	3.92 (d)		
		4-CH	4.18 (q)		
		3-CH	4.25 (t)		
		2-CH	4.38 (t)		
		5-CH	5.95 (d)		
		6-CH	5.97 (d)		
		1-CH	7.87 (d)		
26	Uridine diphosphate (UDP)	CH_2_	4.22 (m)	12.87	23.33
		4-CH	4.28 (m)		
		3-CH	4.37 (m)		
		2-CH	4.37 (m)		
		5-CH	5.98 (d)		
		6-CH	5.99 (d)		
		1-CH	7.96 (d)		
27	Fumaric acid	4,5-CH	5.52 (s)	2.66	1.91
28	α-glucose	1-CH	5.24 (s)	−1.05	−11.43
29	Cytidine	3-CH	6.09 (d)	1.81	2.09
		2-CH	7.87 (d)		
30	Fumarate	CH	6.522 (s)	2.20	2.07
31	Niacinamide	5-CH	7.6 (dd)	−2.28	0.75
		6-CH	8.72 (d)		
32	Adenine	2-CH	8.11 (s)	4.47	4.78
		6-CH	8.12 (s)		
33	Hypoxanthine	2-CH	8.22 (s)	2.31	37.64
		7-CH	8.272 (s)		
34	Formate	CH	8.46 (s)	34	−105.611
35	NADP	6-CH	8.84 (d)	4.54	1.71
		4-CH	9.15 (d)		

Multiplicity: singlet (s), doublets (d), triplets (t), doublet of doublets (dd), multiplets (m); quartets (q).

**Table 2 molecules-24-01841-t002:** Identified differential metabolites.

LPS vs. Control				SPPM60-D vs. Control		
NO.	Metabolites	VIP	Trend	Metabolites	VIP	Trend
1	Lactate	25.36	↓	Lactate	24.06	↓
2	Alanine	14.19	↑	Betaine	12.84	↑
3	Choline	6.61	↑	Proline	7.58	↑
4	Proline	5.46	↑	Creatine phosphate	6.36	↑
5	Betaine	4.27	↑	Creatine	6.14	↑
6	Glycine	4.23	↑	choline	5.90	↑
7	Creatine	4.07	↓	Glycine	5.89	↑
8	Taurine	3.84	↓	Alanine	4.36	↑
9	Acetate	2.45	↑	Taurine	3.21	↓
10	Aspartate	1.82	↑	Acetate	3.10	↑
11	Uridine	1.62	↑	Aspartate	2.81	↑
12	Threonine	1.67	↑	Trimethylamino oxide (TMAO)	2.48	↑
13	Formate	1.51	↓	Isoleucine	2.34	↑
14	Trimethylamine N-oxide (TMAO)	1.43	↓	Threonine	1.23	↑
15	Methanol	1.40	↑	Ethanol	1.05	↓
16	leucine	1.31	↑			
17	Ethanol	1.22	↑			
18	Glutamine	1.13	↓			

**Table 3 molecules-24-01841-t003:** Pathway analysis associated with the differential LPS metabolites.

Name	Total	Expected	Hits	Raw p	−LOG (p)
Glycine, serine and threonine metabolism	31	0.39379	5	2.52E-05	10.589
Methane metabolism	9	0.11433	3	0.00013807	8.8877
Aminoacyl-tRNA biosynthesis	69	0.8765	5	0.001237	6.6951
Glycolysis or Gluconeogenesis	26	0.33028	3	0.003731	5.5911
Nitrogen metabolism	9	0.11433	2	0.0052073	5.2577
Arginine and proline metabolism	44	0.55893	3	0.01646	4.1068
Pyruvate metabolism	23	0.29217	2	0.032939	3.4131
Alanine, aspartate and glutamate metabolism	24	0.30487	2	0.035665	3.3336
D-Glutamine and D-glutamate metabolism	5	0.063514	1	0.062007	2.7805
Cyanoamino acid metabolism	6	0.076217	1	0.073964	2.6042
Pyrimidine metabolism	41	0.52082	2	0.09334	2.3715
Taurine and hypotaurine metabolism	8	0.10162	1	0.097448	2.3284
Primary bile acid biosynthesis	46	0.58433	2	0.11353	2.1757
Valine, leucine and isoleucine biosynthesis	11	0.13973	1	0.13162	2.0278
Glyoxylate and dicarboxylate metabolism	18	0.22865	1	0.20667	1.5766
Glutathione metabolism	26	0.33028	1	0.28493	1.2555
Porphyrin and chlorophyll metabolism	27	0.34298	1	0.29419	1.2235
Glycerophospholipid metabolism	30	0.38109	1	0.32127	1.1355
Valine, leucine and isoleucine degradation	38	0.48271	1	0.38878	0.94474
Purine metabolism	68	0.8638	1	0.58963	0.52826

**Table 4 molecules-24-01841-t004:** Pathway analysis associated with the differential SPPM60-D metabolites.

Name	Total	Expected	Hits	Raw p	−LOG (p)
Glycine, serine and threonine metabolism	31	0.35004	5	1.32E-05	11.232
Glycolysis or Gluconeogenesis	26	0.29358	3	0.0026238	5.9431
Methane metabolism	9	0.10162	2	0.0041113	5.494
Arginine and proline metabolism	44	0.49682	3	0.011797	4.4399
Pyruvate metabolism	23	0.2597	2	0.026347	3.6364
Aminoacyl-tRNA biosynthesis	69	0.77911	3	0.039232	3.2383
Cyanoamino acid metabolism	6	0.067749	1	0.065978	2.7184
Taurine and hypotaurine metabolism	8	0.090332	1	0.087048	2.4413
Primary bile acid biosynthesis	46	0.51941	2	0.092722	2.3782
Nitrogen metabolism	9	0.10162	1	0.097415	2.3288
Glyoxylate and dicarboxylate metabolism	18	0.20325	1	0.18588	1.6827
Alanine, aspartate and glutamate metabolism	24	0.271	1	0.24027	1.426
Glutathione metabolism	26	0.29358	1	0.25762	1.3563
Porphyrin and chlorophyll metabolism	27	0.30487	1	0.26616	1.3236
Glycerophospholipid metabolism	30	0.33874	1	0.29123	1.2336

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
