# Peer review of "1H-NMR Metabolomics Analysis of the Effects of Sulfated Polysaccharides from Masson Pine Pollen in RAW264.7 Macrophage Cells"

_molecules, 2019, doi:10.3390/molecules24091841_

Reviewer 1 Report

This paper describes interesting results, and the results of this study may well be useful to the scientists in the same field. There are, however, a few issues that need to be revised or explained in more detail. Also, all the figures in manuscript have low resolution to read so author should provide the high-resolution of figures. Most of figures are very poor to read their legend of x, y axis and even the statistics.

1.      How did you determine the average molecular weight of SPPM60-D to 40.5 kDa using dextran? The manuscript should be more informative. The molecular weight for SPPM60-D is too big to determine using HPLC. Provide the details for the standard curve data for the various dextran author used to determine the molecular weight for SPPM60-D as supplementary data.

2.      There is no information about PPM60-D in section 2.1. Author should provide the more details for the FT-IR data of PPM60-D and SPPM60-D with their purity.

3.      The legend of X and Y axis for Fig 1 are not written in English, author should stay consistent the language to English. Also, the direction for the allows in figure are not clear.

4.      Author should provide more details how author determined the degree of substitution of SPPM60-D to be 1.212 with relevant references.

5.      In line 72-73. Authors mentioned that no significant differences were noted for 400 and 800 ug/ml of SPPM60-D, but author did not provide the data to compare the groups. In Figure 2, the concentration of S100 to S800 have statistics of p<0.01 compared to control, but no comparison data was provided for S400 and S800.

6.      In Line 83-84, author noted “the concentrations of some of the 84 metabolites differed”, however no details were provided to support the result. Also, Table 1 looks interesting results to support the this result topic, but unfortunately, they are hard to recognize and match with Figure 3.

7.      In line 97, “linear value of 1.35”. no reference to understand the value.

8.      I think that Figure 9 and 10 contains the major subjects for this research, but those data have very poor resolution, so it is very hard to get the point for the research.

Author Response

1.      How did you determine the average molecular weight of SPPM60-D to 40.5 kDa using dextran? The manuscript should be more informative. The molecular weight for SPPM60-D is too big to determine using HPLC. Provide the details for the standard curve data for the various dextran author used to determine the molecular weight for SPPM60-D as supplementary data.

A:The average molecular weight of SPPM60-D was determined by HPLC with Deviating light detector using four dextran standards (Dextran T-10(Mw=10000)、DextranT-70(Mw=70000)、DextranT-110(Mw=110000)、DextranT-500(Mw=500000)). Below is the standard curve of dextrans.

                 standard curve of Dextran

According to the standard curve: logMw = -0.2582t + 11.757 (R2 = 0.951), the average molecular weight of SPPM60-D was MW=40.5 K。

2.      There is no information about PPM60-D in section 2.1. Author should provide the more details for the FT-IR data of PPM60-D and SPPM60-D with their purity.

A: The monosaccharide composition of PPM60-D has showed in section 2.1 :PPM60-D was then purified from PPM60 using Sephacryl S-400HR and was composed of mannose, galactose, trehalose and an unknown monosaccharide at molar ratios of 0.75:1:0.68:2.37 (Geng, 2016) .

 The purity of PPM60-D was detected by Sephacryl S-400HR Gel chromatography. The result was showed in the figure below.

Fig. The results of the PPM60 purification

3.      The legend of X and Y axis for Fig 1 are not written in English, author should stay consistent the language to English. Also, the direction for the allows in figure are not clear.

A: Fig 1 has been revised.

4.      Author should provide more details how author determined the degree of substitution of SPPM60-D to be 1.212 with relevant references.

A: The standard cu ve was made by different concentration of sulfate.

  Concentration(μg/mL)  

Fig. 2-4  The standard curve of the sulfate concentration

The degree of substitution of SPPM60-D and PPM60-Dwere calculated by the formula:DS=(162×S)/(32-102×S).

5.      In line 72-73. Authors mentioned that no significant differences were noted for 400 and 800 ug/ml of SPPM60-D, but author did not provide the data to compare the groups. In Figure 2, the concentration of S100 to S800 have statistics of p<0.01 compared to control, but no comparison data was provided for S400 and S800.

A: There is no significant difference between 400 and 800 ug/ml. P=0.6759

6.      In Line 83-84, author noted “the concentrations of some of the 84 metabolites differed”, however no details were provided to support the result. Also, Table 1 looks interesting results to support the this result topic, but unfortunately, they are hard to recognize and match with Figure 3.

A: We identified 35 metabolites but no 84 metabolites.

7.      In line 97, “linear value of 1.35”. no reference to understand the value.

A: Linear values are data derived from sample modelling and are derived from statistical software statistical analysis.

8.      I think that Figure 9 and 10 contains the major subjects for this research, but those data have very poor resolution, so it is very hard to get the point for the research.

A: We have revised figure 9 and 10.

Reviewer 2 Report

The authors have shown 1H-NMR metabolomics analysis of the effects of sulfated polysaccharides from masson pine pollen in RAW 264.7 macrophage cells.  Many data have been shown and it is interesting.  However, some revisions should be added.  The characters should be bigger in almost all figures.  Chinese was used in Figure.1.  The MTT assay generally indicates cell survival but not proliferation.  If the authors want to show cell proliferation, another approach should be tried.  Top numbers of raw p in Tables 3 and 4 are strange.

Author Response

The authors have shown 1H-NMR metabolomics analysis of the effects of sulfated polysaccharides from masson pine pollen in RAW 264.7 macrophage cells.  Many data have been shown and it is interesting. 

However, some revisions should be added. 

The characters should be bigger in almost all figures. 

A: The characters have been bigger than before in some figures.

Chinese was used in Figure.1. 

A: We have revised.

The MTT assay generally indicates cell survival but not proliferation.  If the authors want to show cell proliferation, another approach should be tried. 

A: We think the MTT assay can detect the survival and proliferation also.

Top numbers of raw p in Tables 3 and 4 are strange.

A: e-05 means the power of -5 times 10